# Is Teleaudiology Achieving Person-Centered Care: A Review

**DOI:** 10.3390/ijerph19127436

**Published:** 2022-06-17

**Authors:** Sophie Brice, Helen Almond

**Affiliations:** 1Department of Nursing and Allied Health Sciences, Faculty of health Arts and Design, Swinburne University of Technology, Hawthorn, VIC 3122, Australia; 2Australian Institute of Health Service Management, College of Business and Economics, University of Tasmania, Hobart, TAS 7000, Australia; helen.almond@utas.edu.au

**Keywords:** digital health, audiology, teleaudiology, person-centered care, intervention

## Abstract

Digital health and person-centered care are unquestionably linked in today’s Australian healthcare landscape. Teleaudiology is the application of digital health in the field of audiology, and it has become a popular component of standard audiological care. Behavior modification is essential in audiology intervention. Guidance on achieving behavior change, which is dependent on digitally enabled intervention, is a valuable resource when used in tandem to achieve person-centered care. The aim of this review is to determine whether teleaudiology achieves person-centered care. A qualitative review was conducted, followed by mapping and analysis. Analysis identified evidence of teleaudiology use, and ascertained guiding principles are appropriate to behavior change dependent digital intervention supported or enabled person-centered care. In conclusion, teleaudiology will continue to be a promising technology for promoting relatedness, a positive user experience, confidence and capability, and appropriate levels of autonomy for the user to choose from among the person-centered care options available.

## 1. Introduction

In today’s Australian healthcare landscape, digital health and person-centered care have become unquestionably linked. As the nation’s lead digital health organization, the Australian Digital Health Agency (ADHA) provides a national digital health strategy and framework for action on digital health [1]. The Australian Commission on Safety and Quality in Health Care proffer the National Safety and Quality Health Service Standards (NSQHSS), which provide essential guidance in achieving high safe quality healthcare, including digital health, and services that emulate the principles of person-centered care [2]. Teleaudiology refers to the use of digital health in the field of audiology [3].

The term teleaudiology is used when a healthcare service specifically refers to hearing services provided by an audiologist or audiometrist who is situated in a different location from the healthcare user [3]. Teleaudiology encompasses a wide range of interactions between a hearing healthcare provider and healthcare users. For example, email, text messaging, phone calls, videoconferencing websites and apps, and web chats. Teleaudiology can occur synchronously (in real time), when an audiology provider interacts with the healthcare user via phone, videoconference, or an app, or asynchronously (with a time delay), when the interaction occurs via mail, email, text messaging, or other means. Alternatively, hearing services can be provided in a hybrid format, in which the healthcare user receives some hearing services in person and others via teleaudiology. 

Teleaudiology has been widely used, globally, for decades [4,5]. The application of teleaudiology has capitalized on technological innovations that have influenced all aspects of intervention in audiological care, from initial assessment [6,7,8] to rehabilitation [6,7,8]. To support the effective use of teleaudiology as part of standard audiological care, Australia’s governing audiology body has recently developed specific guidelines, which are expected to be released in mid-2022 [9]. 

Dr. Carl Rogers’ seminal work in the 1950s established the person-centered care approach as a means for understanding personality and human relationships, and ultimately achieving better health outcomes [10]. Rogers’ field of psychology and the modern field of audiology are linked by the reliance on behavior change. This behavior change reliance underpins chronic disease management in general. Chronic diseases are long-term health conditions influenced by the person experiencing their behavioral choices and actions. An important foundation of person-centered care is the recognition of the importance of supporting behavior change in intervention for chronic disease management [11,12,13,14,15].

Hearing loss is a complex and chronic condition that requires specialized audiological intervention and care. Hearing aid adoption and adherence are behavioral components of audiological intervention. Achieving successful hearing aid adoption and adherence is a well-documented challenge, culminating in high rates of ‘In-the-Drawer’ hearing aids [16,17]. Person-centered care approaches to addressing adoption and adherence include increasing autonomy in hearing aid access through direct-to-consumer provision [18], self -fit hearing aids [19,20], improving rehabilitation support through counseling and coaching programs [21,22], and improving self-management skills [23,24]. The recent release of professional competency standards for audiology in Australia include frequent and explicit application of ‘client-centered care’, which reflects the collective drive to embed person-centered qualities of care into audiological practice [25]. In audiology, the person receiving intervention or audiological care is commonly referred to as a ‘client’, client-centered care in audiology is thus the application of person-centered care in this field. Teleaudiology guidelines currently under development reflect this ideal, with equal emphasis on and application of person-centered principles of care [9].

Teleaudiology is the digitalization of audiological care in practice and reflect the standards of the ADHA framework for action on digital health [1]. In audiology intervention, behavior modification is critical. To achieve person-centered care, guidance on achieving behavior change, dependent on digitally enabled intervention, are valuable resources when used in tandem. In 2015, Yardley et al. published a set of guiding principles to assist with understanding the requirements of behavior changes related to digital health-interventions [26]. They describe their research and intervention development experience as having resulted in the development of a set of person-centered intervention features. These features appear to improve adoption and adherence in most digital interventions. In their conclusion, Yardley et al. [26] present the common guiding principles, applicable to many interventions, as: user autonomy, user competence, and positive emotional experience combined with a sense of relatedness. By emphasizing distinct context-specific behavioral issues, the guiding principles may encourage and enlighten person-centered teleaudiology interventions, and the ultimate goal of hearing aid adoption and adherence.

As the title suggests, the purpose of this review is to understand whether the use of teleaudiology can support person-centered care. The four objectives are to (1) conduct a systematic search of the published and grey literature for the use of teleaudiology in audiological care studies, (2) map out the characteristics and range of audiological intervention tasks used in the identified studies, (3) using Yardley et al. guiding principles [26], assess whether the use of teleaudiology can support person-centered care, and (4) propose recommendations for advancing a person-centered approach and enhancing the consistency of use of person-centered teleaudiology and the way it is reported.

## 2. Materials and Methods

### 2.1. Introduction

The qualitative framework developed by Ritchie et al. served as the foundation for this review methodology [27]. This review began with the formation of a small research team comprised of individuals skilled in digital health research, service delivery and research synthesis. The team collaborated to develop a research question and an overall study protocol, which included identifying search terms and databases to search. The review was divided into five stages: (1) identifying the research question, (2) identifying relevant studies, (3) study selection, (4) data charting, and (5) collating, summarizing, and reporting the findings.

### 2.2. Research Question

This review was guided by the question ‘Is teleaudiology achieving person-centered care?’ For the purposes of this study, a review is described as a qualitative systematic review [27,28].

### 2.3. Data Sources and Search Strategy

The initial search was carried out on 22 April 2022, in three electronic databases: PubMed, EBSCO CINAHL, and Scopus. The databases were chosen for being extensive and in order to cover a wide range of disciplines. The database search was not restricted by date, subject, or type. However, languages other than English were excluded due to a lack of translation resources. The research team’s search query included the following terms: teleaudiology, audiological care, and intervention. The search query was tailored to the specific requirements of each database using Boolean logic (Table 1). Adoption and adherence were chosen as search terms because they are commonly used in the audiological literature pertaining to behavior change with intervention.

### 2.4. Citation Management

All citations were imported into the web-based bibliographic manager software Endnote, Philadelphia, PA, USA, and duplicates were manually removed, with additional duplicates removed as they were discovered later in the process.

### 2.5. Eligibility Criteria

Conscious of the likely paucity of evidence, the research team aimed to identify all studies (original and review) relevant to the research question. Studies that broadly described the use of teleaudiology were eligible for inclusion. Only studies published in English were included. The analysis excluded studies that described teleaudiology without intervention. When the same data were reported in multiple publications (for example, a journal article and an electronic report), only the article with the most complete data set was used.

### 2.6. Title and Abstract Screening

To avoid wasting resources on studies that did not meet the minimum inclusion criteria, only the title and abstract of citations were reviewed for the first level of screening.

### 2.7. Data Characterization

Following title and abstract screening, all citations deemed relevant were obtained for subsequent review of the full-text article. For studies that could not be obtained through the authors’ institutional holdings, attempts were made to interpret the context of the teleaudiology use described. The authors created a form to confirm relevance and extract study characteristics, such as the audiological activities for which teleaudiology was described. Studies were discarded at this stage if they were discovered to be ineligible. The team met after independently reviewing 36 studies to resolve any conflicts and to help ensure consistency between team as well as relevance with the research question and purpose.

### 2.8. Data Summary and Synthesis

The final selection of papers was compiled in the web-based bibliographic manager software Endnote, Philadelphia, PA, USA, and extracted data were then sorted into tables to assist screening for validation and coding. Table 2 identifies a priori phases in which the intervention tasks were completed and offers a description for each phase guided by current audiology professional standards for delivery of care and current practices represented in the literature [25,29]. Data analysis was completed in two stages, using a priori themes and a qualitative framework approach [27]. Stage-one descriptive analysis aided in the ordering and identification of similarities and differences in the qualitative data by creating a matric of intervention tasks as rows charted against the degree of autonomy, which was used to extract data while mapping Yardley et al.’s first principle [26]. Second-order analysis in stage two aided in the development of descriptive and explanatory conclusions that allowed interpretation of the presence of Yardley et al.’s remaining two principles [26].

## 3. Results

### 3.1. Organization and Selection of Reviews

Initially, 3881 studies were identified through database searches. After removing duplicates and incomplete records, 3238 studies remained to be screened. Abstract screening was carried out using Endnote software search tools, with three successive searches using the same criterion, replicating the database searching of the three-criterion described in Table 1. Further screening with Endnote software removed 3202 records, leaving 36 records. The research team individually then together screened these to reach an agreement on the final seven studies to be included for data extraction and analysis (Figure 1).

Data were extracted from seven studies: five research studies [31,32,33,34,35], one review [36] and one case study [37]. The identified studies ranged from 2010 to April 2022, with one per year from 2019 to 2022, when the COVID-19 pandemic began, forcing many audiology clinicians to adopt or consider accepting teleaudiology as part of their potential clinic service offerings [32]. It should be noted that Behl et al. [37] were describing a pediatric case study in the United States. The lack of teleaudiology-specific guidelines in the United States, including in pediatric audiology, which frequently requires specialized guidelines, has an impact on the case study’s use of teleaudiology in the same way that the current lack of local or global teleaudiology guidelines has an impact on all of the articles [38,39]. As a result, application of autonomy can still be objectively identified in accordance with the research question in this review. Ferguson and Henshaw reviewed four related research projects that were not described in any of the other articles [36]. This review was included for the remainder of the analysis due to this and the lack of empirical evidence in the final selection (Table 3).

### 3.2. Descriptive Analysis

Descriptive analysis aided the ordering and identification of similarities and differences in the qualitative data. As presented in Table 4, descriptive analysis assisted in categorizing data into a matrix.

The rows represent the phases of intervention task as described in Table 2 [25,29]. The columns represent the degree of autonomy. The studies were then mapped to the intervention phase and Yardley et al.’s first principle of promoting autonomy [26]. The mapping enabled the presence of Yardley et al.’s remaining two principles of promoting positive experience and relatedness and promoting competence to be interpreted [26]. The description of autonomy according to Yardley et al. is the feeling of being self-directed [26]; the spectrum of autonomy ranges from self-led, the degree in which autonomy is most promoted, to clinician-led, the degree in which autonomy is least promoted.

Yardley et al.’s description of promoting competence is of control and confidence [26]. While autonomy denotes control, supporting competence is derived from tasks that support skills, education, and ability to manage, which is primarily represented by rehabilitation and intervention supportive tasks. Yardley et al.’s third principle, promoting positive experience and relatedness, is based on motivating intentions to participate in the intervention in accordance with affective behavior change ideas [26]. Yardley et al. goes on to explain that intrinsic motivation is achieved by invitation rather than a directive to the user [26]. Qualitative interpretation related to the guiding principle of positive experience or relatedness and competence is thus required based on indication of autonomy, context of intervention task being described, and context derived from the source material.

Data extraction and sorting revealed a wide range of intervention tasks described, with a predominant focus on the rehabilitation and general phases (Table 4). Eikelboom and Swanepoel was the only paper to discuss teleaudiology delivered in various contexts without explicitly detailing an intervention task, as this was a survey detailing how clinicians used the service, with a focus on digital mode of contact [31]. Meyer et al. provided the greatest detail regarding the tasks for which teleaudiology could be used, and this information was collected from people with hearing impairment and their significant others, and as such this paper may be the most representative of actual teleaudiology use [34]. There was evidence of all degrees of autonomy being used across the papers [31,32,33,34,35,36,37], which provided a balance in how much control the person with hearing impairment can be given, but the phases of intervention in which teleaudiology were used were heavily concentrated in the rehabilitation [32,33,34,35,36,37] and general phases [33,34,35,36] (Table 4).

### 3.3. Second Order Analysis

Second order analysis demonstrates the development of the descriptive and explanatory conclusions.

#### 3.3.1. Autonomy

Autonomy, defined as “the feeling self-directed” p.8., is the first guiding principle suggested by Yardley et al. for person-centered application of digital health in behavior-change-dependent intervention [26]. Analysis revealed ten instances of self-directed use of teleaudiology, referred to here as self-led use, across five studies [33,34,35,36]. According to the descriptions used by Eikelboom et al. in reporting on teleaudiology practices, the degree of autonomy was better represented as a spectrum ranging from self-led to clinician-led [29]. Despite self-assessment or self-fit or programming of hearing aids being well established in audiological care [5], there was only one mention of autonomy for the immediate intervention tasks described, and it focused on describing what those with hearing impairment and their support network would like [35]. Taken together, this highlights a gap between methods that have been rigorously validated for common use [29], and yet still evade being common practice [32].

Supporting intervention tasks that require decision making and goal setting were described in a self-led manner by Burden et al. [33], and a shared control or partnership approach by Meyer et al. [34]. Burden et al. [33] also described rehabilitation tasks delivered with teleaudiology in a self-led manner, along with Ferguson and Henshaw [36], with the partnership approach also found for Burden et al. [33] and Behl et al. [37]. Pediatric audiology is governed by specific standards that separate such competencies between treating adults and children. However, it is worth noting that some autonomy for the person and their family was still clearly noted by Behl et al. [37]. More than half of the studies described the provision of information in a self-led format [33,34,35,36]. 

The greatest degree of autonomy was found in clinical tasks that focused rehabilitation and provision of information. The audiological literature encourages information and acceptance prior to beginning intervention, as well as increased support during the rehabilitative phase of intervention, which may be critical to improving rates of adoption and adherence to audiological intervention [40,41,42]. The Australian Audiology competency standards describe person-centered care as a mutually beneficial relationship in which all decisions are shared [25]. According to this standard, the evidence for autonomy and partnership found in supporting intervention and rehabilitation lends support to the use of teleaudiology as a person-centered practice.

#### 3.3.2. Positive Experience or Relatedness

The second guiding principle proposed by Yardley et al. is the promotion of positive experience or relatedness, which is described as increasing users’ perceived relatedness or support from the intervention [26]. According to Yardley et al., respecting the individual’s perspective and contribution, as well as providing tailored and positive feedback where appropriate, is a critical component for achieving this [26]. As a result, this fits into a partnership or shared input approach to supporting intervention and rehabilitation in which the clinician collaborates with the person’s experience to derive and assess outcomes for their audiological care. While all of the studies described rehabilitation-related tasks using teleaudiology [32,33,34,35,36,37], this phase of intervention tasks also showed the greatest balance between self-led and clinician-led delivery. 

#### 3.3.3. Competence

Yardley et al.’s final guiding principle suggests the promotion of competence [26]. This is defined as the concept of control and confidence, in which the individual takes the lead in identifying aspects of their lifestyle to be targeted for behavior change in manageable, minimally disruptive steps. An important consideration is that the changes must be within their technological and financial ability. Yardley et al.’s concept of competence is one of potential competence that is attainable within and through the control of the individual [26]. This differs from autonomy, where the task at hand may be of the person’s design. For example, goal setting dictated by the person, but tasks may also necessitate assistance to accomplish (auditory training in a group session). Competence is important because it is inextricably linked to effective behavior change theories applied in chronic health management, where behavior change is needed to support intervention [12,14]. Only Ferguson and Henshaw [36] explicitly describe competence as a measurable element, identifying it as relevant experience or digital proficiency for using the self-led internet-based intervention program under investigation. This study found no correlation between computer proficiency and program adherence, which supports other findings from audiology research that digital proficiency is not a factor in intervention type selection and commitment [43]. This finding is supported in the wider digital health literature, which points to confidence rather than competence [43,44,45]. Furthermore, the term competence has recently been called into question, with the term “capability” being preferred instead, as it encompasses competence while also recognizing adaptability, continuous learning, and self-efficacy, which supports Yardley et al.’s original intention [46,47,48,49].

The scope of the review did not include a thorough evaluation of whether all teleaudiology tasks promoted capability. The presence of self-led and partnership-led tasks for supporting intervention and rehabilitation, on the other hand, is consistent with the goals of this principle. Ferguson and Henshaw [36] described training programs that were available for the individual to use in their own time, manageable according to their chosen time available, and as such supported self-efficacy and manageable behavior change. Burden et al. [33] explained an online interactive decision coaching guide created through participatory design with people who have hearing loss. Along with Yardley et al.’s second principle, the program itself supports capability [26]. Meyer et al. [34] conducted a study that found that computer and technology access, general use, and application in hearing loss intervention were found for a group of people with hearing loss and their significant others, which strongly supports the third guiding principle. The cohort’s technology demographic showed variation that could be representative of the wider Australian population, but what piqued our interest was a strongly positive result on the eHealth Literacy Scale for those with hearing impairment and their significant others (Table 4 of Meyer et al.) [34]. The participants in Meyer et al. identified that the tasks for which they used technology were mostly related to appointments and information seeking via modes of internet searching, phone, email, and apps (Figure 1 of Meyer et al.) [34]. The study of Meyer et al. clearly demonstrates capability, but the breakdown of the results they share demonstrates low response rates for apps (*n* = 7) and videoconferencing (*n* = 21) compared to basic telephone (*n* = 64) and email (*n* = 81) that are less supportive of self-efficacy [34]. Given the rise of app-based teleaudiology accessibility in almost all major hearing aid user apps that now accompany hearing aids, this may speak more to hearing care professionals’ well-documented skepticism to accept and adopt teleaudiology for supporting user hearing care capability, which we seek to support [32,50,51].

## 4. Discussion

The purpose of this review was to identify teleaudiology services literature that facilitated person-centered evidence-based practices. The three guiding principles proposed by Yardley et al. [26] were chosen as success criteria for this review due to their relevance to behavior change dependent interventions, which in audiology require hearing aid adoption and adherence. A thorough literature search was conducted, yielding seven studies describing the use of teleaudiology as an intervention process for audiological calls from 2010 to 2022. All seven studies found evidence in support of Yardley et al.’s three guiding principles, indicating that teleaudiology can support or enable person-centered care.

### 4.1. Implications for Audiological Care

Globally, teleaudiology is being integrated into standard audiological practice. While there is clear evidence that teleaudiological methods have been validated, there is no clear answer as to whether teleaudiology should be considered as a complement or auxiliary part of clinical audiological best practice. Infrastructure constraints have been recognized as an important factor that will determine possible and appropriate teleaudiology use [51,52]. In addition to practical needs, clinician preferences that will influence adoption and use of teleaudiology have been identified, which may act as a barrier of access for clients who are offered teleaudiology as part of their hearing care [51,53]. To date, the clinical preference in clinical care has been for hybrid teleaudiology, allowing clinical judgement to prevail where infrastructure and personal factors such as lack of training or suitability may impact appropriate and effective teleaudiology in clinical care provision [51]. Underuse of teleaudiology services in clinical practice has been recognized globally, even after the pressures of the recent pandemic. This underuse highlights that cross-sectional and possibly global efforts to support use and implementation of teleaudiology are required [32,51,54].

The creation of specific guidelines for teleaudiology by Australia’s professional audiology body is evidence of the changes in recognizing teleaudiology as an important part of standard audiological care [9,25]. The differences in jurisdiction regulations for providing, reimbursing and supporting audiological care with teleaudiology will shape the application and practice-based learnings. Recognition of common objectives, outcomes and best practice methods across various regions using teleaudiology will be a potentially effective path towards global collaborative recognition of best practice knowledge and methods in teleaudiology.

The evidence collected in this review was predominantly research. The case study by Behl et al. [37] and the research survey conducted by Meyer et al. [34] were the only two articles that described a retrospective account of teleaudiology services that included or was led by the client’s perspective. Eikelboom and Swanepoel [32] and Parmer et al. [31], on the other hand, presented the clinician’s perspective of teleaudiology services delivered, with the remaining articles describing the use of teleaudiology as part of research initiatives. What is difficult to determine from this array is how representative this evidence is of actual teleaudiology use in commercial as well as clinical settings. Given the industry’s strong theoretical emphasis on person-centric, partnership-focused, shared decision-making competency requirements in Australia [9,25], the relatively low representation of client-led perspective in the evidence found here is a concern for determining whether there is a challenge of representation between practice-based evidence and praxis. In future, co-design research methods, which include communities of practice should be considered as contributors to research-based evidence to inform best-practice methods in modern audiological care.

While there is evidence to support each of Yardley et al.’s three guiding principles, confirming that teleaudiology can support or enable person-centered care, there are observations that can be used to inform teleaudiology’s future application.

#### 4.1.1. Autonomy

In the audiological literature, autonomy has been discussed under two main themes: self-fit and self-management. A self-fit hearing aid is one that allows the user to complete all aspects of the intervention process on their own. Two examples of self-fit hearing aids are: the Blamey Saunders Hears hearing aid range [19], which has been available in Australia for over a decade, and a more recent competitor, Bose, in America [20], which required a change to American federal regulation [55]. The Blamey Saunders Hears hearing aid range was introduced in 2009 as a hybrid service model, whereby users could seek as much or as little clinical support as they desired, in person or via teleaudiology [56]. The use of teleaudiology to complement the service model has also been recognized as a digitally enabled model of care by the ADHA [1]. Yardley et al.’s third guiding principle of competence (or, more accurately, capability), in which the user has confidence and competence to gain control over their intervention, is also complemented by the ability to choose the level of assistance sought or provided. Despite a well-established commercial presence that continues to grow, this review found no evidence of self-fit hearing aid systems or digitally enabled models [18,43,57,58]. Similarly, self-led hearing assessment has been present for several years [6,7,8], but it was not identified in this review either. Potentially limited evidence was generated as a result of a significant mismatch between practices that do not generate evidence in the literature and research that does represent commercial or clinical practice. There is an incongruence between theory and practice. As a result of this gap, developing effective evidence-based practices will be hampered while relevant evidence remains difficult to obtain.

There is broader audiological literature available that can contribute to the understanding of the impact of autonomy in supporting success in behavior change dependent intervention in audiology. Humes et al. [59] indicated that autonomy over how users accessed hearing aids, for example, clinical or teleaudiology-based, had a powerfully positive effect on improving adoption and adherence of hearing aids. The Humes et al. [59] study was significant because it was conducted in a field where adoption and adherence are the best available outcome measures for determining whether or not suitable behavior change has been achieved with intervention, and thus far have not been very successful [16,17]. Yardley et al. contend that promoting autonomy has the potential to improve outcomes [26]. Factors influencing the ability to successfully use autonomy cannot yet be elucidated from the limited literature available and are beyond the scope of this study. One common suggestion is that age is inversely related to digital proficiency. Given the older average age of the hearing-impaired population and preliminary evidence that digital proficiency alone is not indicative of likely success with teleaudiology services, the field would benefit from further research into success factors of teleaudiology services supporting autonomy [45,48]. The primary goal of person-centered care is to improve health outcomes, particularly for chronic diseases such as hearing loss [10,12,14]. According to preliminary evidence gathered by Australia’s audiology governing body, the use of teleaudiology as part of care delivery can improve health outcomes even further [53]. The authors recommend that the field of audiology strive to better understand how to apply autonomy as part of person-centered care, using teleaudiology, in order to combat the limited successes of adoption and adherence seen so far.

#### 4.1.2. Positive Experience and Relatedness

Yardley et al. [26] defines the promotion of users’ perceived relatedness or support from the intervention as one of positive experience and relatedness. Outcome measures in audiology have a long history of using what is primarily user-led goal setting and decision making, with the two most common forms being the Consumer-Oriented Scale of Improvement [60] and the Glasgow Hearing Aid Benefit Profile [61], both of which are over 20 years old. Apart from Behl et al. [37], who discussed audiological care for pediatrics, whose needs differ from those of adults, it is, therefore, fitting that the studies that did describe goal-setting and decision-making tasks did so in a partnership or self-led manner [3,4,5,6]. Recent trends in audiological research include the use of Ecological Momentary Assessment because it enables more realistic and personalized evaluation and goal setting, particularly where teleaudiology allows for contextually honest and valid input from the individual [62,63], which cannot effectively be captured in clinic-based care [64]. In the future, there will be more opportunities for audiological care to use teleaudiology to promote relatedness and intervention support, allowing for person-centered care.

#### 4.1.3. Competence

Yardley et al. [26] defined the final guiding principle of control and confidence as competence; however, the authors argue that capability is a more appropriate term. Manageable tasks that are in line with the person’s lifestyle, i.e., causing minimal disruption to their current routine, in order to promote successive, hopefully successful, steps towards the intervention’s required behavior modification. The definition of teleaudiology provided by Australia’s governing body for audiology practice includes counseling and education, implying that supporting audiological services can also be accessed asynchronously, i.e., the user accesses and uses the resources independently of the clinician [3]. Dr. Mel Ferguson has performed outstanding work in providing person-led education and training programs for people with hearing loss, with encouraging results in their adherence to intervention [36,65,66,67]. One of these studies was identified by the review described in this study [36]. Among the studies examined, rehabilitation and general access to information were the most strongly identified, supporting the use of teleaudiology for these parts of the intervention process in a person-centered manner. Control over access to necessary information or programs, with the goal of instilling confidence in the individual to apply their learnings in achieving behavior change, is a clear application of the principles of self-efficacy explained by Dr. Albert Bandura in his seminal work on health behavior principles [13]. Given the removal or reduction of clinical appointment time and administrative constraints, the ability to provide access to resources in a delivery mode that is amenable to the person’s lifestyle and capacity to manage new information is fundamentally advantageous using teleaudiology. The need to triage potential clients based on their needs and align greater resources to assist them in achieving self-efficacy and successful and sustained behavior change itself is an important concept raised in the audiological literature [42,67,68]. Even if the person undergoing intervention has full autonomy over all aspects of the intervention process, the principles of person-centered care should remind us that confidence and general capability are also strongly supported alongside control.

### 4.2. Learning from the Broader Literature of Digital and Health

Person-centered care has emerged as a prominent theme in the literature on both audiological care and digital and health. Person-centered care is explicitly advocated for in frameworks and standards developed by the ADHA and the Australian Commission on Safety and Quality of Health Care Services (NSQHSS) [1,2]. Other domains, such as culture and literacy, are also considered in the delivery of high-quality care.

#### 4.2.1. Culture

The Australian Digital Health Agency and the National Quality and Safety Health care Standards both recognize culture as an important component of governance and leadership [1,2]. Workforce planning and clinical culture are two outcomes of culture that the audiological field could improve upon. The Australian Audiology standards do not recognize culture in the same context, instead emphasizing professional safety, learning, and behavior across three of the six domains listed [25]. Despite repeated evidence of clinical efficacy, the use and adoption of teleaudiology in audiological care has been clearly documented as being limited by clinician attitudes and reluctance [50,51,52,53,54]. This review revealed a small number of studies describing the application of teleaudiology in intervention. This is surprising given the long history of teleaudiology use [4,5], as well as a somewhat moderate increase in use due to the COVID-19 pandemic limiting how many clinicians can continue to practice [51,53]. The results suggest that professional paternalism towards practice exists [69]. Clinical barriers are significant impediments to the effective adoption and implementation of teleaudiology. Partnership between the person and the clinician us essential for an effective healthcare delivery relationship, which is best represented as a meeting of a care expert and an experience expert [70]. The authors advocate for taking culture, particularly professional culture, into account in order to foster a safe and supportive environment enable use of teleaudiology towards person-centered care.

#### 4.2.2. Literacy

Health literacy is also a strong area of focus in Australia’s digital and health literature, with the goal of supporting effective partnerships [1,2]. The audiology standards address this need by emphasizing the clinician’s responsibility to adapt to the persons current level of digital health literacy, as well as the responsibility to develop and maintain their professional digital literacy capability fir safe and quality audiological practice [25]. This is a promising start, and it is supported by the studies examined in this review in terms of using participatory design or demonstrating successful use of internet programs despite prior computer technology experience [34,36]. This contrasts with the historical issue of poor health literacy among users and ineffective literacy standards regarding resources provided, according to the literature [71,72]. Meeting digital literacy standards may be a more effective path to achieving and delivering a person-centered focus on all literacy levels for the user.

## 5. Conclusions

Teleaudiology has grown to unprecedented levels of use and acceptance in audiological care, undoubtedly aided by the recent COVID-19 pandemic. As audiology’s digital healthcare delivery application, it is critical to ensure that teleaudiology can maintain rigorous validated care standards not only as a healthcare service, but also as a digital service that aspires to emulate person-centered care. This review identified and analyzed evidence of teleaudiology use, determining guiding principles appropriate to behavior change dependent digital intervention supported or enabled person-centered care. 

The identified literature was deemed insufficient to fully represent the current use of teleaudiology in audiological intervention, potentially leading to a mismatch between evidence-based practice. This has implications for developing evidence-based practices that should effectively leverage existing field learnings. Autonomy in audiology is visible in available teleaudiology applications, but there was little evidence of it in immediate intervention. To ensure the field’s future success in adoption and adherence, it is suggested that the field of audiology work to better understand how autonomy across audiological care with teleaudiology can support person-centered care. Teleaudiology applications are constantly evolving and improving methods for delivering different aspects of the audiology intervention process.

Teleaudiology will continue to be a promising tool for promoting relatedness and positive user experience, confidence and capability, and appropriate levels of autonomy for the user to choose from among the person-centered care options available. Observations from other fields, such as health and digital domains, culture, and literacy, can provide opportunities to supplement teleaudiology and ensure person-centered practice as part of an effective holistic audiological healthcare partnership.

## 6. Limitations

The strength of evidence included is a limitation of this review. Due to the scarcity of evidence identified, the decision was made not to exclude evidence of lower ranking strength (case study, expert opinion). However, in order to answer the research question, it was necessary to reflect current practices rather than recommendations. Evidence of current practices is required to inform changes or renewal of recommendations for teleaudiology, a rapidly evolving field of health care services.

Another limitation was the global disparity in teleaudiology practice, as well as the varying degrees to which teleaudiology is recognized as a component of standard audiological care. Teleaudiology-specific guidelines are about to be published in Australia. At the time of writing this review, the authors were unaware of any other incorporation of teleaudiology into the professional practice guidelines published by audiological governing body, though there are independent reports and guidelines that can be sourced. Due to the lack of regional or global teleaudiology guidelines, it is not possible or appropriate to evaluate the use of teleaudiology. Each jurisdiction will have different local regulatory and infrastructure influences that cannot be equitably compared. It may be too early to conduct this review, but it is also important to recognize that if teleaudiology is practiced, there should be evidence, research, and the development of best-practice guidelines to ensure the highest standards of care are maintained.

The final limitation is the selection of search terms. While the most common terms used in audiological literature were chosen, there is still the possibility of missing articles that did not use common terms. For example, ‘adoption’ is a universally accepted term for the user’s acceptance of intervention, such as hearing aids, which are almost the only intervention available; however, the term ‘uptake’ can also be found. Sustained behavior change in audiological care is dependent on the adoption and adherence of intervention, which is typically hearing aids, and thus the use of search terms that are broad in terms of language but specific in terms of clinical application resulted in a relatively small pool of articles identified for the research question posed here. Future research may have to choose between refining the search terms at the expense of leaving too little to be identified or expanding the criteria beyond the scope of the research question, i.e., accuracy vs. validity.

## Figures and Tables

**Figure 1 ijerph-19-07436-f001:**
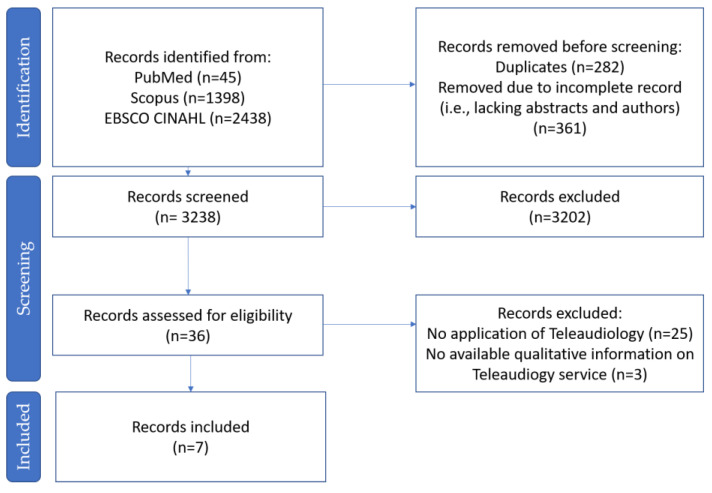
PRISMA 2020 flow chart (reprinted/adapted with permission from Liberati et al [30]. 2020, S. Brice).

**Table 1 ijerph-19-07436-t001:** Search terms as criterion applied as variable terms to identify studies pertaining to the use of teleaudiology in audiological intervention practices.

Criterion	Variable Term 1	Variable Term 2	Variable 3
1	Teleaudiology	Telehealth	Telepractice
	Tele-audiology	Tele-health	Tele-practice
2	Hearing loss	Audiology	-
3	Intervention	Adoption	Adherence

**Table 2 ijerph-19-07436-t002:** Phase and intervention task descriptions.

Phase	Intervention Tasks	Description
Immediate intervention	Assessment/testing/screening	Typically, intervention begins with assessment of a hearing loss. This may be testing or screening (short version of a diagnostic test), followed by fitting of a hearing device (usually a hearing aid but may also refer to the fitting process of a cochlear process which occurs after implantation surgery), and finally adjustment or programming of the hearing device to improve the sound based on the user’s feedback
	Fitting (of a hearing aid/device/cochlear implant)	
	Adjustment/programming (post fitting)	
Supporting intervention	Decision making	Decision making and goal setting are applied in the early phase of intervention to support intervention acceptance, outcomes measurements and engagement
	Goal Setting	
Rehabilitation	Training (auditory/communication/tinnitus)	All intervention is supported by a degree of rehabilitative support, even in the absence of fitting hearing devices. Auditory training and communication strategies are commonly addressed to support the person adjust to their intervention, a degree or counseling or coaching to address expectations management is also common and usually continues long after the fitting of a hearing device for weeks or months until the user can continue independently
	Counseling/coaching	
	Communication strategies	
General	Information/education (content can include maintenance and handling guidance)	Throughout audiological intervention requires a degree of information sharing. The topics could cover maintenance or handling of hearing aids, or aid learning of communication strategies or provide guidance on any of the tasks described so far

**Table 3 ijerph-19-07436-t003:** Included publications.

Paper	Publication	Type	Content	Research Data Capture
Behl et al. [37]	Exceptional Patient Magazine	Article	Case study	United States
Burden et al. [33]	American Journal of Audiology	Research	Participatory Design	United States
Eikelboom and Swanepoel [31]	American Journal of Audiology	Research	Research Survey	International
Ferguson and Henshaw [36]	American Journal of Audiology	Review	Research Forum	U.K.
Meyer et al. [35]	Ear and Hearing	Research	Group Concept Mapping	Australia
Meyer et al. [34]	Perspectives of the ASHA Special Interest Groups	Research	Research Survey	Australia
Parmar, Beukes and Rajasingam [32]	International Journal of Audiology	Research	Mixed Methods cross sectional survey	U.K.

**Table 4 ijerph-19-07436-t004:** Studies selected following screening and eligibility analysis with qualitative analysis according to audiological intervention process, tasks, and degree of autonomy described.

Phase	Degree of Autonomy				
	Self-Led	Shared Control/Partnership	Clinician-Led	Identified but not Described	Not Identified
Immediate intervention	Meyer et al. [34]		Meyer et al. [34] Parmar, Beukes and Rajasingam [32]		Behl et al. [37] Burden et al. [33] Eikelboom and Swanepoel * [31] Ferguson and Henshaw [36]
Supporting intervention	Burden et al. [33]	Meyer et al. [34]	Behl et al. [37]		Behl et al. [37] Eikelboom and Swanepoel * [31] Ferguson and Henshaw [36] Meyer et al. [35] Parmar, Beukes and Rajasingam [32]
Rehabilitation	Ferguson and Henshaw [36] Burden et al. [33]	Behl et al. [37] Burden et al. [33]	Meyer et al. [34] Parmar, Beukes and Rajasingam [32] Behl et al. [37]	Meyer et al. [35]	Eikelboom and Swanepoel * [31]
General	Burden et al. [33] Ferguson and Henshaw [36] Meyer et al. [34] and Meyer et al. [35]	Meyer et al. [34] Meyer et al. [35]	Meyer et al. [34]		Behl et al. [37] Eikelboom and Swanepoel * [31] Parmar, Beukes and Rajasingam [32]

* The digital mode of teleaudiology delivery, e.g., call, email, or video-call, was described rather than the tasks performed.

## Data Availability

Not applicable.

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
