# Peer review of "Is Teleaudiology Achieving Person-Centered Care: A Review"

_ijerph, 2022, doi:10.3390/ijerph19127436_

Round 1

Reviewer 1 Report

The present manuscript is well written and focuses an interesting and current subject. The academic structure is adequate. The different sections allow a pleasant and easy reading. The manuscript is well organized and complete, covering aspects that are very relevant to the topic under analysis.

I want to congratulate the authors on this way of presenting their study.

However, several points as indicated below need to be addressed by authors to improve the quality of the article:

1. Despite being a review article, I propose that the abstract has highlighted the different parts of the article's structure: Introduction, objective, material and methods, results and conclusion.

2. They should clarify whether in the eligibility criteria only original articles were considered or review articles were also considered.

3. You mentioned: Data were extracted from seven studies, six research studies, and one case study (line 168). In this type of analysis, can the case study not bias the results?

4. Is the user's autonomy not related to their age and general health status? This type of variables were not analyzed. It would be interesting to put a table with the characterization of the sample and instruments used in each of the articles included in this review

5. In the future, will teleaudiology be able to replace face-to-face practice? Which model is better, knowing that there are positive and negative aspects to both.

Author Response

Please see the attachment for the breakdown of response and actions to feedback provided.

Reviewer 2 Report

In the text authors mention "... The research team’s search query included the following terms: teleaudiology, audiological care, and intervención..."  but un table 1 they describe "hearing AND loss", in the same table "adoption and adherence" are broad terms. 

Authors initially state "The database search was not restricted by date, language, subject, or type" and after ".. Studies published in languages other than English were excluded due to a lack of translation resources.....", please be more precise. 

Some phrases are ambiguous, for example ".... The most descriptive paper was one that detailed the use of teleaudiology by people with hearing impairment and their significant...... ", which one? 

and so throughout the text. 

I consider that the review must be shortened and describe in a succinct form. There are repetitive information and some ambiguous paragraphs that must be corrected. What about a comparison of teleaudiology in several countries, nothing is mention about it. 

Author Response

(The authors gave the same response as above.)

Reviewer 3 Report

- In the introduction at the beginning of the 3rd line, an {f} appears before the word Australian which should be deleted

- The article carries out a qualitative review of the literature that addresses the topic of teleaudiology. The review of teleaudiology and analyzed the implementation of its principles, of digital systems in audiological care.

- Is a review article that summarizes the state of the art on the topic.

- This article summarizes and presents the different guidelines used in different countries, allowing the creation of new guidelines with the good practices of each country. This could allow the creation of broader guidelines for the whole world through international audiology associations.

- The study was well designed, using the best databases of scientific literature

- The conclusions of the study study are in line with the reality and the state of the art of this topic. Teleaudiology was already carried out before, but with the emergence of the Covid19 pandemic, the need for a better implementation of teleaudiology became clearer and, on the other hand, the pandemic also made professionals and users more accepting of this type of intervention.

- References are adequate and up-to-date, but a bit extensive. 

- The tables presented are easy to read and very didactic.

Author Response

(The authors gave the same response as above.)

Round 2

Reviewer 2 Report

No comments